# Gentiopicroside-Induced gastric cancer necroptosis via the HIF-1 signaling pathway: A study involving molecular docking and experimental validation

Bo Xiong[1⊙], Mingjie Fan[2⊙], Zhihui Wang[1], Xiaolu Yang[1], Shan Cao[1], Jie Shen[3‡*], Beibei Fan[1‡*]

1 Department of Clinical Pharmacy, Baoshan Hospital Affiliated to Shanghai University of Traditional Chinese Medicine, Shanghai, China, 2 Department of Pharmacy, Shanghai Fourth Rehabilitation Hospital, Shanghai, China, 3 Department of Pharmacy, Shuguang Hospital Affiliated to Shanghai University of Traditional Chinese Medicine, Shanghai, China

⊙ These authors contributed equally to this work.
‡ BF and JS also contributed equally to this work.
* jessiefan2012@163.com (BF); shj421@126.com (JS)

**Data Availability Statement:** All relevant data are within the paper and Supporting Information files.

## Abstract

### Objectives

Gentiopicroside is an effective treatment for several types of cancer, inducing numerous forms of programmed cancer cell death. However, there are few investigations into the role of necroptosis. By utilizing molecular docking, and experimental validation, this study aims to investigate whether gentiopicroside elicits necroptosis in gastric cancer.

### Methods

Using software PyMOL and AutoDock, gentiopicroside was docked with RIPK1, RIPK3, MLKL and HIF-1α proteins. And a cell study was performed based on SGC7901 cells. The necroptosis-related proteins and HIF-1 signaling pathways were explored using western blot (WB) analysis. Finally, an animal study was performed to test the inhibitory effect in vivo.

### Results

Docking studies indicated that the docking energies of gentiopicroside to necroptosis-related proteins and necroptosis-characteristic proteins are all below -5 kcal/mol. Additionally, gentiopicroside cells reduce gastric cancer viability and inhibit proliferation. Results from the animal experiments indicated that gentiopicroside inhibits the growth of the gastric cancer xenograft tumor. Western blot and immunohistochemistry (IHC) staining demonstrated that gentiopicroside higher p-receptor-interacting protein kinase 3(p-RIPK3) levels in vitro and in vivo.

**Funding:** This work was supported by [Shanghai Baoshan District Science and Technology Commission medical health project] (Grant numbers [21-E-52]), the Excellent Young Medical Talents Training Program and National nature cultivation fund project of Shanghai Baoshan District Hospital of Integrated Traditional Chinese and Western Medicine(No.: 2022BY008) and the National nature cultivation fund project of Shanghai Baoshan District Hospital of Integrated Traditional Chinese and Western Medicine (No.: GZRPYJJ-201805). The funders had no role in study design, data collection and analysis, decision to publish, or preparation of the manuscript.

**Competing interests:** The authors have declared that no competing interests exist.

## Conclusion

The findings of this study revealed that necroptosis is involved in the inhibitory effect of gentiopicroside toward gastric cancer.

## Introduction

Necroptosis is a novel form of programmed cell death (PCD) closely associated with several types of tumors that destroys cellular structures and stimulates cell death. Necroptosis is unique because there is no detectable caspase activity, which is typical of PCD apoptosis. Additionally, necroptosis is driven by a signaling cascade involving receptor-interacting protein kinase 1 (RIPK1), receptor-interacting protein kinase 3 (RIPK3), and the pseudokinase mixed lineage kinase domain-like protein (MLKL) [1]. These proteins are commonly recognized as markers of necroptosis [2].

Growing evidence suggests that necroptosis-related genes play crucial roles in gastric cancer cell function, including proliferation and migration in vitro [3]. The recently developed necroptosis-related gene prognostic index (NRGPI) is a proven prognostic biomarker that screens gastric cancer patients with a cold tumor immune microenvironment [4]. The NRGPI effectively predicts prognosis and immunotherapy efficiency in cases of gastric cancer by measuring AXL, RAI14, and NOX4 levels [5]. Furthermore, a necroptosis risk model based on several genes (NPC1L1, GAL, RNASE1, PCDH7, NOX4, GJA4, SLC39A4, BASP1, BLVRA, NCF1, PNOC, and CCR5) was validated on patients with gastric cancer [6]. As a result, necroptosis is now widely employed in gastric cancer management. Several new compounds and drugs inhibit gastric cancer growth in vitro and in vivo through necroptosis. For instance, benzophenanthridine alkaloid chelerythrine is a novel inhibitor that targets thioredoxin reductase and promotes gastric cancer cell necroptosis [7]. Besides, gambogic acid also inhibits gastric cancer cell growth through necroptosis [8]. Consequently, necroptosis has emerged as a new PCD target of gastric cancer.

Gentiopicroside is one of the important iridoid components and the main active component in Gentiaceae. Recent studies have confirmed that gentiopicroside has various pharmacological effects such as inhibiting inflammation [9], anti-apoptosis [10] and inhibiting tumor cell proliferation [11]. It can be seen that gentiopicroside is an effective biomolecule with many important biological activities and therapeutic significance. Studies have shown that gentiopicroside has good application potential for gastric mucosal injury [12] and gastric cancer [13], while necrotic apoptosis can be activated in inflammatory diseases and cancer [14]. HIF-1 has been recognized as an important anticancer drug target [15]. Accumulating evidence indicates that HIF-1α may be a potential driver of Hypoxia-induced HIF-1α/lncRNA-PMAN inhibits ferroptosis by promoting the cytoplasmic translocation of ELAVL1 in peritoneal dissemination from gastric cancer in gastric cancer [16]. The activation of HIF-1α can also promote necrotic apoptosis [17]. Therefore, we reasoned that gentiopicroside can induce necrotic apoptosis of gastric cancer through HIF-1 signaling pathway. This study investigates the role of necroptosis in the treatment of gastric cancer with gentiopicroside via molecular docking and experimental validations in vitro and in vivo.

## Materials and methods

### Reagents and chemicals

Standard gentiopicroside was purchased from Standard Corporation (Shanghai, China). Primary and secondary antibodies were procured from Abcam (MA, USA). The Cell Counting

Kit-8 (CCK-8) and biochemical kits for oxidative stress levels and pro-cytokine levels were provided by Beyotime (Nantong, Jiangsu, China). Dulbecco's Modified Eagle Medium (DMEM), fetal bovine serum (FBS), and Annexin V & Dead Cell Reagent Assay Kit were purchased from Merck (MO, USA).

## Molecular docking

The 3D-structure pdb format files of RIPK1 protein (PDB ID: 4ITH), RIPK3 protein (PDB ID: 7MON), MLKL protein (PDB ID: 4MWI) and HIF-1α protein (PDB ID: 2ILM) were downloaded from the Protein Data Bank (PDB) website [18]. Besides, the gentiopicroside structure was obtained from PubChem. Next, ChemBio3D version 14.0 software was used to minimize the binding energy and convert it into a 3D structure. The data were saved in mol2 format files. RIPK1, RIPK3, MLKL and HIF-1α were set as receptors, while gentiopicroside was set as a ligand. The AutoDockTools-1.5.6 with Autodock 4.1 program package was employed to generate the ligand and perform the hydrogenation of proteins. AutoDock Vina [19] was used for calculating the binding energy of the molecular docking. The docking results were visualized with PyMoL 2.4.1 [20].

## Cell culture and group assignment

Gastric cancer SGC7901 cells were obtained from the cell bank of the Chinese Academy of Sciences (Shanghai, China) and cultured in DMEM with 10% FBS. The culture method was carried out according to American Type Culture Collection (ATCC) guidelines [21]. When the gastric cancer SGC7901 cell density reached 80%, the cells were randomly divided into five groups: (1) Control group (Control); (2) Gentiopicroside at 4 mg/mL group; (3) Gentiopicroside at 8 mg/mL group; (4) Gentiopicroside at 16 mg/mL group; (5) Gentiopicroside at 16 mg/mL combined with Necrostatin-1(Nec-1) group (16 mg/mL+Nec-1). Nec-1 (10 μM), a necroptosis inhibitor, was added 2 hours before the experiments were conducted [22].

## Cell counting kit-8 assay

CCK-8 kits were used to determine cell viability [23]. Briefly, gastric cancer SGC7901 cells were cultured in 96-well plates and treated with rising concentrations of gentiopicroside. The CCK-8 solution was added 48 h later and absorbance at 450 nm was measured using a microplate reader (PT-3502, Ponetov, Beijing, China).

## Colony formation assay

Gastric cancer SGC7901 cells were seeded in 6-well plates, then cultured and treated with gentiopicroside. Different doses of GDC-0326 were added following previously published methods [24,25]. When visible colonies formed, the colonies were washed, fixed, and stained with 0.05% crystal violet in 20% ethanol. The criterion was that more than 50 cells counted as a valid colony, as per previous reports [24,25]. Cell proliferation was calculated as: clone formation rate (%) = [colonies counted / seeded cells] × 100% [24,25].

## Flow cytometry assay

Gastric cancer SGC7901 cells were treated with gentiopicroside. After 48 h, the gastric cancer SGC7901 cells were collected. A dye solution was added and the mixture was incubated in the dark for 20 min according to manufacturer instructions. Finally, the fluorescence intensity of the SGC7901 cells at 488/530 nm was analyzed using a flow cytometer (FACSCalibur) [26]. The necroptosis ratio was recognized in the Q2 zone, following existing literature [27].

## Western blot

The total protein content of cells was isolated and the concentrations were measured with a bicinchoninic acid assay kit. Equal amounts of protein were separated onto gels and transferred to polyvinylidene difluoride (PVDF) membranes. Consequently, the PVDF membranes were blocked using a blocking solution, then the membranes were incubated with diluted primary antibodies overnight. The following day, the PVDF membranes were incubated with secondary antibodies for 2 h. All proteins were visualized by enhanced chemiluminescence reagent and the relative densities of the proteins were normalized to housekeeping proteins using ImageJ software.

## Animal model establishment and group assignment

Approval for the animal study was obtained from the Laboratory Animal Ethics Committee of the Shanghai University of Traditional Chinese Medicine (PZSHUTCM2308260006). BALB/C nude mice (all male with weights of 19–20 g) were provided by Shanghai Jie-si-jie Laboratory Animal Co., Ltd. (Shanghai Lab Animal Grant Number: SCXK (H) 2023–0004). The mice were at specific pathogen-free (SPF) grade and housed strictly following animal welfare guidelines. The temperature was 20–25˚C, the humidity was 40±5%, and a 12 h light/ dark cycle was implemented. The mice were randomly assigned into five groups (n = 6): (A) Control group; (B) 12 mg/kg group; (C) 24 mg/kg group; (D) 48 mg/kg group; (E) Cisplatin (DDP) group. Gastric cancer SGC7901 cells were planted into the mice following established procedures. The tumor sizes were recorded using the following formula: length × width 2/2 [28,29]. When the tumor size reached 100 $mm^3$, gentiopicroside was administered at 0.4 mL per day, in different gentiopicroside concentrations, according to the group. Positive group mice were given DDP injections three times per week (Monday, Wednesday, and Friday) at 2 mg/kg [30]. In the control group, 0.4 mL of PBS was administered orally once per day. The tumor sizes were recorded every three days between tumor sizes of 100 $mm^3$ to approximately 1000 $mm^3$.

The standard for the humane endpoint is set to euthanize all mice in all groups when the average volume of the tumor in the model group is greater than 1000 mm3. Mice died of cervical dislocation after anesthesia with pentobarbital sodium (40 mg/kg, i.p). The Animal Ethics Committee reviewed and agreed on the setting of humane endpoints and the method of execution. Throughout the animal experiment, the Laboratory Animal Ethics Committee checked the experimental process and ensured that the handling of experimental mice complied with relevant welfare policies.

## Immunohistochemistry assay

To observe the morphology and investigate related protein expression after gentiopicroside administration, the tumors were subjected to immunohistochemistry (IHC) analysis. The tumors were carefully collected to avoid contamination from other tissue. The tumor samples were then cut, blocked, and incubated with primary antibodies (1:150, ab179800, Abcam). The next day, tumor slides were incubated with secondary antibodies (1:150, ab179800, Abcam). Images of the tumor samples were then obtained with a microscope (MF31, Mshot, Guangzhou, China). Relative intensity was calculated using integrated optical density (IOD) data and the area of immunostaining was determined using NIH Image-Pro Plus 6.0 software (Media Cybernetics Co., MD, USA).

## Statistical analysis

Results from the animal experiments were expressed as mean ± SD and presented by Graph-Pad Prism version 9.5 (GraphPad Software Inc., San Diego, CA, USA). Statistical analysis was

**Table 1. Binding energies of compounds to proteins.**

| Compound | energies (kcal/mol) | | | | |
|---|---|---|---|---|---|
| | **HIF-1α (4H6J)** | **RIPK1(4ITH)** | **RIPK3(7MON)** | | **MLKL(4MWI)** |
| Gentiopicroside | -5.7 | -6.4 | -7.3 | | -7.8 |

performed using a one-way ANOVA with SPSS statistical software version 27.0 (IBM, Chicago, IL, US). P < 0.05 was considered statistically significant.

## Results

### Molecular docking study

The gentiopicroside was subjected to molecular docking to confirm their binding to RIPK1, RIPK3, MLKL and HIF-1α proteins. The docking results are shown in Table 1, which indicates that ligand gentiopicroside was closely linked to the RIPK1, RIPK3, MLKL and HIF-1α proteins. All the docking energies were lower than -5 kcal/mol, which reflects the relative stability among RIPK1, RIPK3, MLKL, HIF-1α proteins, and gentiopicroside, as per existing criteria [31]. Fig 1 illustrates that gentiopicroside was bound to the active pocket surface of RIPK1, RIPK3, MLKL, HIF-1α proteins. As shown in Fig 1A, the gentiopicroside-HIF-1α complex is principally stabilized by hydrogen bond with Leu248 and Thr288, the pi-alkyl and pi-sigma interaction with Tyr276 amino acid residues. And the gentiopicroside-RIPK1 conjugate was established by hydrogen bond with Thr110,Tyr255 and Leu104, the alkyl interaction with Leu112 and Pro111 amino acid residues, while also interacting via carbon hydrogen bonding with residues Ser109 and Glu254 (Fig 1B). Furthermore, gentiopicroside-RIPK3 conjugate was established by hydrogen bond with residues Thr94, Met97, Asp160 and Leu92, while also interacting via the alkyl with residues Val35, Ala48, Val27 and Leu149 (Fig 1C). Additionally, gentiopicroside-MLKL conjugate is principally stabilized by hydrogen bond with Lys230, Lys331, Arg333 and Ser335, while also forming carbon hydrogen bond with residue Asn336 (Fig 1D). In conclusion, the molecular docking results unveiled potential grafting sites between gentiopicroside and RIPK1, RIPK3, MLKL and HIF-1α proteins.

### Effect of gentiopicroside on gastric cancer SGC7901 cell viability

As Fig 2A indicates, gentiopicroside administration reduced the cell viability of gastric cancer SGC7901 cells in a dose-dependent manner. Furthermore, when the Z-VAD-FMK inhibitor was added to the high-dose gentiopicroside group, the cell viability was significantly lower than in the high-dose group, suggesting the existence of another type of cell death. When Nec-1 was given to the high-dose gentiopicroside group, the cell viability was higher than in the high-dose group, implying that the other type of cell death was necroptosis. The colony assay exhibited a similar tendency, where gentiopicroside reduced the colony numbers, as Fig 2B shows. However, a combination of Nec-1 and gentiopicroside promoted colony formation more significantly than with gentiopicroside alone.

### Effect of gentiopicroside on gastric cancer SGC7901 cell death

The flow cytometry results are displayed in Fig 2C, which reveals that the necroptosis cells in the upper left quadrant were higher in gentiopicroside than in the control group. Experiments showed that exposure to gentiopicroside serum led to a remarkable but dose-dependent increase in the necroptosis cell ratio in gastric cancer SGC7901 cells compared with the control group.

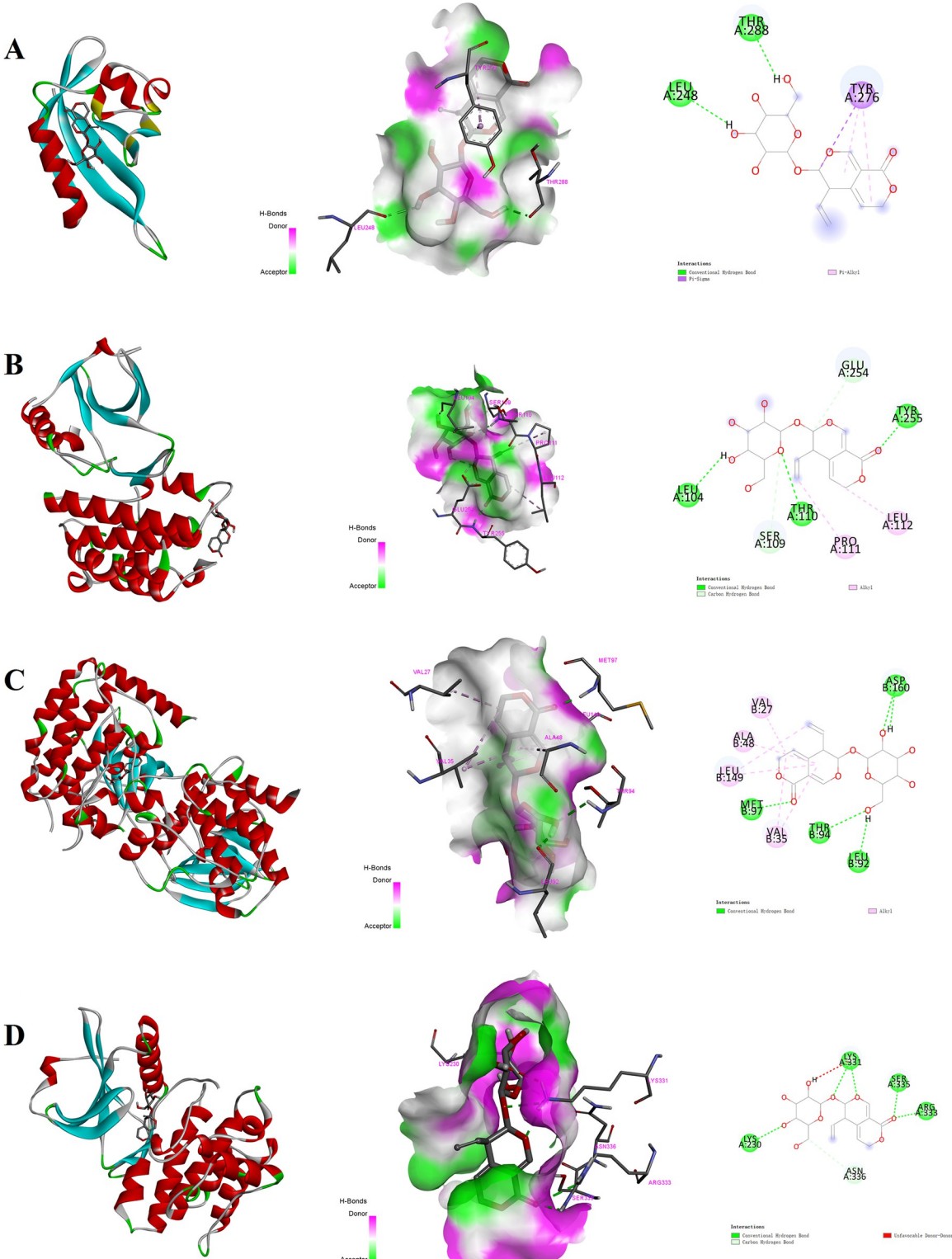

**Fig 1.** Two and three-dimensional images of compound Gentiopicroside docking with RIPK1, RIPK3, MLKL and HIF-1α protein: (A) Gentiopicroside docking with HIF-1α; (B) Gentiopicroside docking with RIPK1; (C) Gentiopicroside docking with RIPK3; (D) Gentiopicroside docking with MLKL.

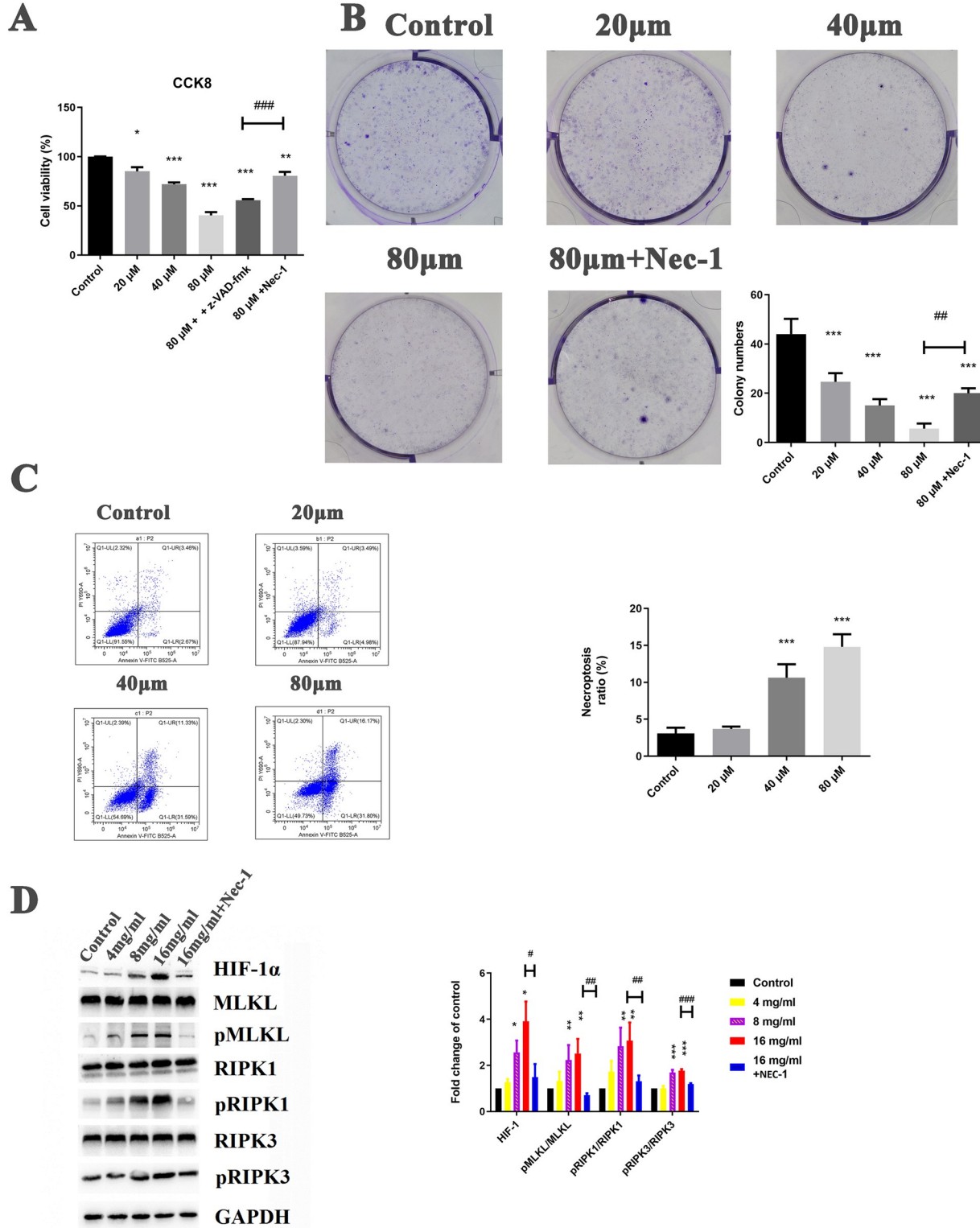

**Fig 2. Cell experiments.** (A) Cell viability assay using CCK-8; (B) Colony assay and quantification; (C) Flow cytometry assay; (D) Western blot experiments and quantification. Experiments were repeated three times. Statistical differences compared with the model group were considered significant at * P < 0.05, ** P < 0.01, *** P < 0.001.

### Effect of gentiopicroside on necroptosis-related proteins in vitro

Results of the western blot analysis in Fig 2D reveal that gentiopicroside firstly up-regulated HIF-1α levels. Furthermore, the WB analysis shows an increase in p-RIPK1/RIPK1 protein levels in the gentiopicroside group over the control group. Gentiopicroside enhanced p-RIPK3/RIPK3 protein levels and p-MLKL/MLKL protein levels, suggesting the activation of necroptosis in gastric cancer SGC7901 cells.

### Effect of gentiopicroside on tumor size and weight in vivo

Fig 3A reveals that body weight was higher in the tumor model group but it decreased after gentiopicroside administration. Fig 3B shows a similar trend regarding tumor size, especially in the several observations made before euthanization. The tumors at the time of euthanization are displayed in Fig 3C, which shows that the xenograft tumor weights were lower after gentiopicroside administration than in the xenograft model group.

### Effect of gentiopicroside on necroptosis and proliferation protein activity in vivo

Results of the IHC assay, which are shown in Fig 3D, reveal that after gentiopicroside administration, the HIF-1α levels were higher than in the xenograft model group. Moreover, as Fig 3D indicates, the p-RIPK3 levels were higher in the three gentiopicroside groups than in the xenograft model group.

## Discussion

As a novel type of regulated cell death, necroptosis is being increasingly applied in cancer treatment. This study investigates the regulating role of the bioactive compound gentiopicroside during necroptosis in gastric cancer cases.

Necroptosis is widely recognized as a novel inflammation-induced PCD pathway when the regular apoptotic pathway is inhibited. Therefore, necroptosis is classified as an alternative apoptotic PCD. Necroptosis is initiated by activation of the serine/threonine RIPK1. Many cytokines that the immune system of the human body encounters elicit the activation of necroptosis, such as the tumor necrosis factor (TNF) family of death receptors, interferon (IFN) receptors, and toll-like receptors (TLRs) [32]. In TNF-induced necroptosis, RIPK1 is activated and additional reactive oxygen species (ROS) are produced in the mitochondria [33]. Subsequently, RIPK1 forms a protein complex (necrosome) with RIPK3 activation through its RIP homologous interaction motif (RHIM) domains. In IFN-induced necroptosis, RIPK3 is activated by the RHIM-containing protein ZBP1 but without the help of RIPK1. Consequently, RIPK3 phosphorylates the substrate MLKL, elicits MLKL oligomerization, and translocates to the plasma membrane.

Necroptosis can be induced by several signaling pathways, such as the HMGB1/TLR4 [34], RIPK3-CaMKII-mPTP [35], AMPK [36], and JNK signaling pathways [37]. HIF-1α is an important signaling pathway that responds to low oxygen in the human body and commonly provokes necroptosis. HIF-1α modulates the tricarboxylic acid cycle and pentose phosphate pathway, resulting in macrophage apoptosis, necroptosis, and a reduction in autophagy [38]. HIF-induced necroptosis plays an important role in colorectal cancer cells [39]. Besides, the up-regulation of HIF-1α in macrophages induces necroptosis by mitochondrial dysfunction, modulated by miR-210 and miR-383 [40]. HIF-1α also participates in necroptosis in ischemic brain injuries [41].

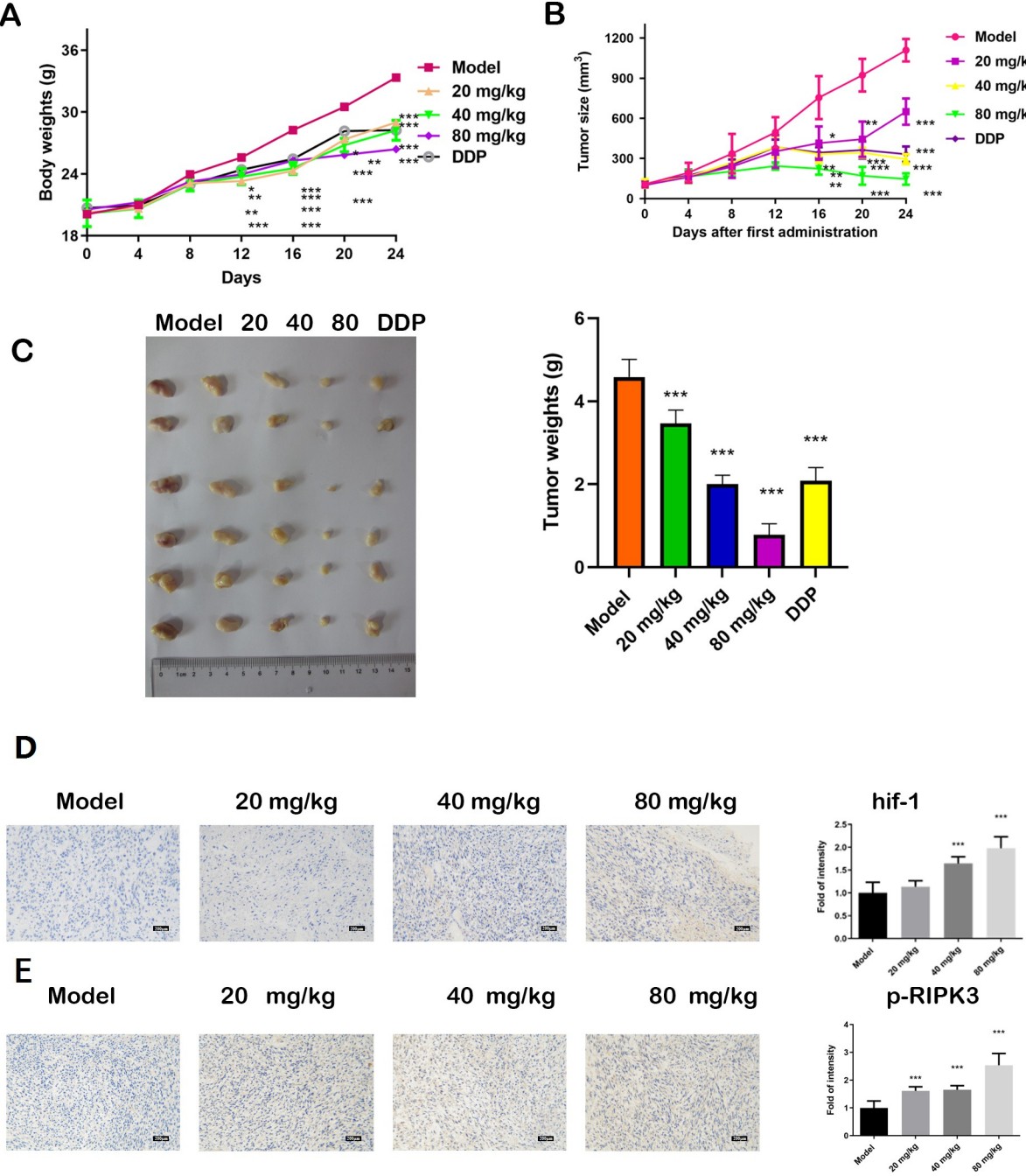

**Fig 3. Animal experiments (# 6 in each group).** (A) Tumor sizes; (B) Tumor weights; (C) Tumor properties at the time of euthanization; (D) Representative HIF-1 IHC and quantification. Magnification = 200; Scale bar: 200μm. (E) Representative p-RIPK3 IHC and quantification. Magnification = 200. Scale bar: 200μm. Statistical differences compared with the model group were considered significant at * P < 0.05, ** P < 0.01, *** P < 0.001.

In recent years, scholars have discovered that many natural products and related agents inhibit cancer cell proliferation by necroptosis. The multi-target molecular mechanism of the marine compound brugine against breast cancer was predicted partly through necroptosis [42]. Additionally, the benzophenanthridine alkaloid chelerythrine stimulates the necroptosis of gastric cancer cells [7]. Gentiopicroside is a bio-active nature-derived compound from the

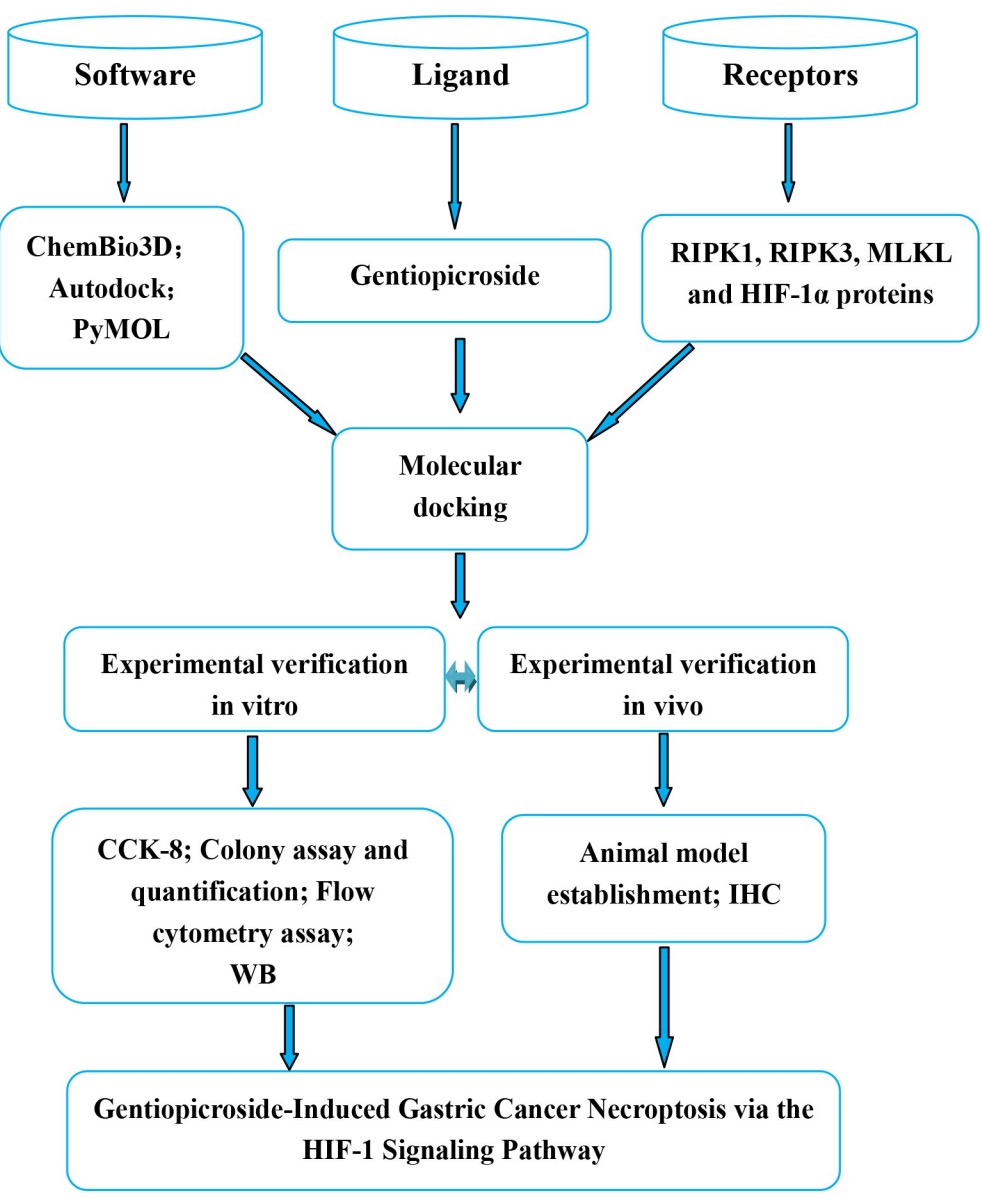

**Fig 4. Workflow of Gentiopicroside-Induced necroptosis in gastric cancer via the HIF-1 pathway.**

genus Gentiana of the family Gentianaceae. Gentiopicroside is found in *Gentiana macrophylla Pall* [12], *Gentiana straminea Maxim* [43], *Gentiana scabra Bunge* [44], and *Gentiana robusta* King *ex Hook. f.* [45]. In recent years, gentiopicroside has been recognized as an effective anti-cancer agent that inhibits the proliferation of numerous types of cancer cells in vitro and in vivo [46], including cervical cancer [47], gastric cancer [13], and ovarian cancer [11]. In gastric cancer, gentiopicroside inhibits the proliferation of gastric cancer by regulating the EGFR/PI3K/AKT signaling pathway [48]. In this study, gentiopicroside was found to induce necroptosis-related protein expression in gastric cancer SGC7901 cells by modulating the HIF-1 signaling pathway.

This study does have some limitations. This is a preliminary study on gentiopicroside participates in the necroptosis of gastric cancer cells. In the present study, we have only studied

SGC-7901 cells. Some studies indicate that the level of HIF-1 protein in SGC-7901 cell line is variable according to different hypoxia situation [49]. The hypoxia situation effect might complicate tumor model result. Some gastric cancer cell line which is not sensitive to oxygen fluctuation to investigate gentiopicroside effect on HIF-1 protein level and related molecular mechanism have not been studied by us, but this does not hamper the value of our research. We will focus on the above issues in the future.

## Conclusion

Through molecular docking, and in vitro and in vivo experiments, this study demonstrated that gentiopicroside participates in the necroptosis of gastric cancer cells. Gentiopicroside inhibits gastric cancer proliferation by inducing necroptosis through the HIF-1 signaling pathway.

The summary diagram illustrating the workflow involved in experimental study of Gentiopicroside-Induced Necroptosis in Gastric Cancer via the HIF-1 Pathway (Fig 4).

## Supporting information

**S1 Raw data.**
(ZIP)

**S1 Raw images.**
(PDF)

## Author Contributions

**Conceptualization:** Jie Shen, Beibei Fan.

**Data curation:** Bo Xiong, Mingjie Fan, Xiaolu Yang, Shan Cao.

**Formal analysis:** Zhihui Wang.

**Funding acquisition:** Mingjie Fan.

**Investigation:** Bo Xiong, Mingjie Fan, Zhihui Wang.

**Methodology:** Bo Xiong.

**Project administration:** Jie Shen, Beibei Fan.

**Supervision:** Jie Shen, Beibei Fan.

**Validation:** Xiaolu Yang, Shan Cao.

**Writing – original draft:** Jie Shen.

**Writing – review & editing:** Beibei Fan.

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
