## [Decision Letter · Decision Letter 0]

11 Jul 2024

PONE-D-24-22481Gentiopicroside-Induced Gastric Cancer Necroptosis via the HIF-1 Signaling PathwayPLOS ONE

Dear Dr. Fan,

Thank you for submitting your manuscript to PLOS ONE. After careful consideration, we feel that it has merit but does not fully meet PLOS ONE’s publication criteria as it currently stands. Therefore, we invite you to submit a revised version of the manuscript that addresses the points raised during the review process.

We look forward to receiving your revised manuscript.

Kind regards,

Jian Hao

Academic Editor

PLOS ONE

“The authors would like to thank the Excellent Young Medical Talents Training Program and National nature cultivation fund project of Shanghai Baoshan District Hospital of Integrated Traditional Chinese and Western Medicine(No. : 2022BY008; No.: GZRPYJJ-201805).”

5. We note that your Data Availability Statement is currently as follows: [All relevant data are within the manuscript and its Supporting Information files.]

Additional Editor Comments:

Docking studies are not accepted unless followed by benchwork assessing affinity. A proposed mechanism of action is required. The compound you identified is a small molecule and these may well show non specific ‘docking’ effects. This should be addressed. Further evidence is needed to ascertain the scientific plausibility of this claim.

Reviewers' comments:

Reviewer's Responses to Questions

**Comments to the Author**

1. Is the manuscript technically sound, and do the data support the conclusions?

Reviewer #1: Partly

Reviewer #2: Partly

Reviewer #3: Yes

2. Has the statistical analysis been performed appropriately and rigorously? 

Reviewer #1: No

Reviewer #2: Yes

Reviewer #3: Yes

3. Have the authors made all data underlying the findings in their manuscript fully available?

Reviewer #1: Yes

Reviewer #2: No

Reviewer #3: Yes

4. Is the manuscript presented in an intelligible fashion and written in standard English?

Reviewer #1: Yes

Reviewer #2: No

Reviewer #3: Yes

5. Review Comments to the Author

Reviewer #1: The manuscript tried to identify the function of gentiopicroside in inducing gastric cancer necroptosis and tried to obtain the possible mechanism via identifying the right pathway. Firstly, the authors applied in silicon method to find the gastric cancer-associated protein targeted by gentiopicroside. And the authors not only find the targeted protein but also calculated the detail binding possibility through molecular docking method. Then, the authors studied the function of gentiopicroside in SGC7901 cell and found that gentiopicroside induce the gastric cancer cell necroptosis through upregulating HIF-1α. Further, the author applied the animal experiment and proved such function of gentiopicroside in vitro. This work is written well and the results proved the potential of gentiopicroside in gastric cancer therapy. The manuscript is recommended to be accepted by the journal Plos One. However, there are some concerns for the authors:

1) In the Introduction section, the authors wrote this section too generally about gentiopicroside. The author may need to rewrite this section especially a sole paragraph about gentiopicroside should be added.

2) In the 1st sentence of the section 2.1, the author should revise this sentence using the research object as the subject. Same to the 1st sentence of section 3.1.

3) In section 2.2, the version of cytoHUbba plugin should be given.

4) In section 2.6, one reference should be added about the ATCC guideline.

5) In section 2.10, the version of ImageJ software should be given.

6) In section 2.11, the full name of DDP should be given.

7) In section 3.2, the result of molecular docking is very good. However, the authors should give more detail information especially the potential chemical group.

8) For Figure 4D, rulers should be added to the pictures.

9) The authors should clarify the parallel number of experiments for each legend of each Figure.

10) In the Discussion section, the authors mentioned the previous work that “gentiopicroside inhibits the proliferation of gastric cancer by regulating the EGFR/PI3K/AKT signaling pathway”. How the authors exclude the possibility of gentiopicroside in inducing gastric cancer necroptosis through the same patyway?

11) A summary diagram needs to be added and cited in the discussion section.

Reviewer #2: Regarding this manuscript, I have the following concerns that need to be conveyed to the authors, and I hope to receive detailed responses:

1, The potential targets list of all docking results are not shown in the manuscript so the strength of relationship between gentiopicroside and RIPK1 as well as HIF-1 proteins remains elusive;

2, Since docking method is crucial to finding the targets, the authors need to clearly describe their docking method;

3, In Figure 3D, the p-MLKL western blot imaging is too ambiguous to conclude the quantification result;

4, Since HIF-1 protein level is very variable according to different hypoxia situation in SGC-7901 cell line (10.1007/s13277-014-1928-7). The authors need to rule out the possibility that HIF-1 level increasing is caused by hypoxia situation change but not from gentiopicroside treatment;

5, The authors need to apply another gastric cancer cell line which is not sensitive to oxygen fluctuation to investigate gentiopicroside effect on HIF-1 protein level and related molecular mechanism;

6, Reviewer has similar concern about that the hypoxia situation effect might complicate tumor model result;

7, The introduction part of this manuscript does not clearly describe the background of current research and also does not highlight the significance of this research.

Reviewer #3: The study is well-structured and comprehensive, addressing the effects of gentiopicroside on gastric cancer through a multi-faceted approach that includes network pharmacology, molecular docking, and both in vitro and in vivo experimental validation. The integration of network pharmacology with molecular docking and biological experiments provides a robust framework for understanding the mechanistic pathways through which gentiopicroside affects gastric cancer cells. The use of both in vitro (cell lines) and in vivo (xenograft models) experiments helps validate the biological relevance and therapeutic potential of gentiopicroside. Investigating necroptosis as an alternative pathway for inducing cell death in cancer cells is promising, especially for forms of cancer where apoptosis might be dysregulated.

Potential Improvements:

While the study does a good job at identifying interactions and effects, deeper mechanistic insights could be achieved. For instance, it could further investigate how gentiopicroside influences other cellular pathways that might be interacting with the necroptosis pathway.

While SGC7901 cells are a valid model for gastric cancer, including additional gastric cancer cell lines could help validate the findings across different genetic backgrounds and enhance the generalizability of the results.

In vivo studies should also explore the long-term effects of gentiopicroside treatment on overall health and survival, as well as potential toxicity, to better understand its feasibility as a therapeutic agent.

6. PLOS authors have the option to publish the peer review history of their article (what does this mean?). If published, this will include your full peer review and any attached files.

Reviewer #1: No

Reviewer #2: No

Reviewer #3: No

---

## [Author Response · Author response to Decision Letter 0]

10 Sep 2024

Reponses to Additional Editor (original comments by editors are in blue color)

1.Comment: Docking studies are not accepted unless followed by benchwork assessing affinity. A proposed mechanism of action is required. The compound you identified is a small molecule and these may well show non specific ‘docking’ effects. This should be addressed. Further evidence is needed to ascertain the scientific plausibility of this claim.

1. Reply: Thanks for pointing this out. After careful consideration, we found some problems in the section of network pharmacology. In order to maintain the rationality of our article, we decided to delete the section of network pharmacology analysis, and relevant content has been modified accordingly. At the same time, we have revised the title of the article: Gentiopicroside-Induced Gastric Cancer Necroptosis via the HIF-1 Signaling Pathway: a study involving molecular docking and experimental validation. Please review the revised draft for details.

Reponses to reviewers(original comments by reviewers are in blue color)

Reviewer#1:

1.Comment: In the Introduction section, the authors wrote this section too generally about gentiopicroside. The author may need to rewrite this section especially a sole paragraph about gentiopicroside should be added.

1.Reply: Thank you for your comment. Your suggestion is valuable. We have rewrited the Introduction section and added a sole paragraph about gentiopicroside. The specific contents are as follows:

“Gentiopicroside is one of the important iridoid components and the main active component in Gentiaceae. Recent studies have confirmed that gentiopicroside has various pharmacological effects such as inhibiting inflammation [9], anti-apoptosis [10] and inhibiting tumor cell proliferation [11]. It can be seen that gentiopicroside is an effective biomolecule with many important biological activities and therapeutic significance. Studies have shown that gentiopicroside has good application potential for gastric mucosal injury [12] and gastric cancer [13], while necrotic apoptosis can be activated in inflammatory diseases and cancer [14]. HIF-1 has been recognized as an important anticancer drug target [15]. Accumulating evidence indicates that HIF-1α may be a potential driver of Hypoxia-induced HIF-1α/lncRNA-PMAN inhibits ferroptosis by promoting the cytoplasmic translocation of ELAVL1 in peritoneal dissemination from gastric cancer in gastric cancer[16]. The activation of HIF-1α can also promote necrotic apoptosis [17]. Therefore, we reasoned that gentiopicroside can induce necrotic apoptosis of gastric cancer through HIF-1 signaling pathway.”

The references are as follows:

[9] Xie X, Li H, Wang Y, Wan Z, Luo S, Zhao Z, et al. Therapeutic effects of gentiopicroside on adjuvant-induced arthritis by inhibiting inflammation and oxidative stress in rats. Int Immunopharmacol. 2019; 76: 105840. doi:10.1016/j.intimp.2019.105840.

[10] Lian LH, Wu YL, Wan Y, Li X, Xie WX, Nan JX. Anti-apoptotic activity of gentiopicroside in D-galactosamine/lipopolysaccharide-induced murine fulminant hepatic failure. Chem Biol Interact. 2010; 188(1): 127-133. doi:10.1016/j.cbi.2010.06.004.

[11] Li X, Yang C, Shen H. Gentiopicroside exerts convincing antitumor effects in human ovarian carcinoma cells (SKOV3) by inducing cell cycle arrest, mitochondrial mediated apoptosis and inhibition of cell migration. J BUON. 2019; 24(1): 280-284. PMID: 30941981.

[12] Yang Y, Wang Z, Zhang L, Yin B, Lv L, He J, et al. Protective effect of gentiopicroside from Gentiana macrophylla Pall. in ethanol-induced gastric mucosal injury in mice. Phytother Res. 2018; 32(2): 259-266. doi:10.1002/ptr.5965.

[13] Huang Y, Lin J, Yi W, Liu Q, Cao L, Yan Y, et al. Research on the Potential Mechanism of Gentiopicroside Against Gastric Cancer Based on Network Pharmacology. Drug Des Devel Ther. 2020;14:5109-5118. doi:10.2147/DDDT.S270757.

[14] Frank D, Vince JE. Pyroptosis versus necroptosis: similarities, differences, and crosstalk. Cell Death Differ. 2019; 26(1): 99-114. doi:10.1038/s41418-018-0212-6.

[15] Masoud GN, Li W. HIF-1α pathway: role, regulation and intervention for cancer therapy. Acta Pharm Sin B. 2015; 5(5): 378-389. doi:10.1016/j.apsb.2015.05.007.

[16] Lin Z, Song J, Gao Y, Huang S, Dou R, Zhong P, et al. Hypoxia-induced HIF-1α/lncRNA-PMAN inhibits ferroptosis by promoting the cytoplasmic translocation of ELAVL1 in peritoneal dissemination from gastric cancer [published correction appears in Redox Biol. 2022 Sep;55:102402. doi: 10.1016/j.redox.2022.102402]. Redox Biol. 2022; 52: 102312. doi:10.1016/j.redox.2022.102312.

[17] Karshovska E, Wei Y, Subramanian P, Mohibullah R, Geißler C, Baatsch, I.,et al. HIF-1α (Hypoxia-Inducible Factor-1α) Promotes Macrophage Necroptosis by Regulating miR-210 and miR-383. Arterioscler Thromb Vasc Biol. 2020; 40(3): 583-596. doi:10.1161/ATVBAHA.119.313290.

2.Comment: In the 1st sentence of the section 2.1, the author should revise this sentence using the research object as the subject. Same to the 1st sentence of section 3.1.

2.Reply: Thank you for your comment. After careful analysis, we found some problems in the section of network pharmacology. In order to maintain the rationality of our article, we decided to delete the section of network pharmacology analysis, and relevant content has been modified accordingly. Our article focuses on molecular docking and experimental studies in vivo and in vitro. Please refer to the revised draft for details.

3.Comment: In section 2.2, the version of cytoHUbba plugin should be given.

3.Reply: Thank you for your comment. We are using Cytoscape version 3.8.2 and the cytoHubba plugin is one of the plug-ins in this version. After careful analysis, we found some problems in the section of network pharmacology. In order to maintain the rationality of our article, we decided to delete the section of network pharmacology analysis, and relevant content has been modified accordingly. Our article focuses on molecular docking and experimental studies in vivo and in vitro. Please refer to the revised draft for details.

4.Comment: In section 2.6, one reference should be added about the ATCC guideline..

4.Reply: We have added one reference about the ATCC guideline. As follows: 

[21] Ferenczy MW, Major EO. Contamination of SVG p12 cells with BK polyomavirus occurred after deposit in the American Type Culture Collection. J Virol. 2014; 88(21): 12928-12929. doi:10.1128/JVI.01600-14.

5.Comment: In section 2.10, the version of ImageJ software should be given.

5.Reply: The version of ImageJ software is ImageJ V1.8.0.112. But, after our deep thinking, we decided to delete the section of network pharmacology analysis, and relevant content has been modified accordingly. Our article focuses on molecular docking and experimental studies in vivo and in vitro. Please refer to the revised draft for details.

6.Comment: In section 2.11, the full name of DDP should be given.

6.Reply: The full name of DDP is Cisplatin. DDP is short for cisplatin.

7.Comment: In section 3.2, the result of molecular docking is very good. However, the authors should give more detail information especially the potential chemical group.

7.Reply: Thank you for your comment. After careful analysis, we found some problems in the section of network pharmacology. In order to maintain the rationality of our article, we decided to delete the section of network pharmacology analysis, and relevant content has been modified accordingly. Our article focuses on molecular docking and experimental studies in vivo and in vitro. Please refer to the revised draft for details. 

8.Comment: For Figure 4D, rulers should be added to the pictures.

8.Reply: Thanks for pointing out our problem. We have added a Scale bar in Figure 4D、E. The modified picture is as follows:

9.Comment: The authors should clarify the parallel number of experiments for each legend of each Figure.

9.Reply: All experiments were repeated three times. We have added the parallel number of experiments for each legend of each Figure. 

10.Comment: In the Discussion section, the authors mentioned the previous work that “gentiopicroside inhibits the proliferation of gastric cancer by regulating the EGFR/PI3K/AKT signaling pathway”. How the authors exclude the possibility of gentiopicroside in inducing gastric cancer necroptosis through the same pathway?

10.Reply: Thank you for your comment. Chen Q, et al have studied how GPS regulates the EGFR/PI3K/AKT signaling pathway under in vitro and in vivo conditions use AGS and HGC27 cells. The activation of HIF-1α can also promote necrotic apoptosis. This pathway is different from our study. Necroptosis is driven by a signaling cascade involving RIPK1, RIPK3, and the MLKL proteins which are commonly recognized as markers of necroptosis. The activation of HIF-1α can promote necrotic apoptosis. Our study demonstrated that gentiopicroside inhibits gastric cancer proliferation by inducing necroptosis through the HIF-1 signaling pathway. 

11.Comment: A summary diagram needs to be added and cited in the discussion section.

11.Reply: We have added a summary diagram detailing the workflow involved in experimental study of Gentiopicroside-Induced Necroptosis in Gastric Cancer via the HIF-1 Pathway in the discussion section. 

Reviewer#2:

1.Comment: The potential targets list of all docking results are not shown in the manuscript so the strength of relationship between gentiopicroside and RIPK1 as well as HIF-1 proteins remains elusive.

1.Reply: Thank you for your comment. After careful analysis, we found some problems in the section of network pharmacology. In order to maintain the rationality of our article, we decided to delete the section of network pharmacology analysis, and relevant content has been modified accordingly. Our article focuses on molecular docking and experimental studies in vivo and in vitro. Gentiopicroside was docked with RIPK1and HIF-1 proteins. Please refer to the revised draft for details.

2.Comment: Since docking method is crucial to finding the targets, the authors need to clearly describe their docking method.

2.Reply: Thank you for your comment. We have already described the content related to molecular docking methods in the molecular docking section of the article, and added a part of the content in the maniscript.

3.Comment: In Figure 3D, the p-MLKL western blot imaging is too ambiguous to conclude the quantification result.

3.Reply: Thank you for pointing out our problem. After checking, the picture is really ambiguous. We have changed the picture and now it is clearer.

4.Comment: Since HIF-1 protein level is very variable according to different hypoxia situation in SGC-7901 cell line (10.1007/s13277-014-1928-7). The authors need to rule out the possibility that HIF-1 level increasing is caused by hypoxia situation change but not from gentiopicroside treatment.

4.Reply: Your suggestion is valuable. We are aware of this, relevant content has been added to the discussion, as follows:

This study does have some limitations. This is a preliminary study on gentiopicroside participates in the necroptosis of gastric cancer cells. In the present study, we have only studied SGC-7901 cells. Some studies indicate that the level of HIF-1 protein in SGC-7901 cell line is variable according to different hypoxia situation[49]. The hypoxia situation effect might complicate tumor model result. Some gastric cancer cell line which is not sensitive to oxygen fluctuation to investigate gentiopicroside effect on HIF-1 protein level and related molecular mechanism have not been studied by us, but this does not hamper the value of our research. In the near future, we will focus on the above issues.

Reference: 

[49] Miao ZF, Zhao TT, Wang ZN, Xu YY, Mao XY, Wu JH, et al. Influence of different hypoxia models on metastatic potential of SGC-7901 gastric cancer cells. Tumour Biol. 2014; 35(7): 6801-6808. doi: 10.1007/s13277-014-1928-7.

5.Comment: The authors need to apply another gastric cancer cell line which is not sensitive to oxygen fluctuation to investigate gentiopicroside effect on HIF-1 protein level and related molecular mechanism.

5.Reply: Your suggestions are of great value to us. We have noticed that as well. We have added this part to the discussion section of the article, which will also be the direction of our future research. In the future, we will select some gastric cancer cell lines that are not sensitive to oxygen fluctuations to study the effect of gentiopicroside on HIF-1 protein level and related molecular mechanism. The relevant content has been added to the document.

6.Comment: Reviewer has similar concern about that the hypoxia situation effect might complicate tumor model result.

6.Reply: Your suggestion is valuable. We are aware of this, relevant content has been added to the discussion, as follows:

This study does have some limitations. This is a preliminary study on gentiopicroside participates in the necroptosis of gastric cancer cells. In the present study, we have only studied SGC-7901 cells. Some studies indicate that the level of HIF-1 protein in SGC-7901 cell line is variable according to different hypoxia situation[49]. The hypoxia situation effect might complicate tumor model result. Some gastric cancer cell line which is not sensitive to oxygen fluctuation to investigate gentiopicroside effect on HIF-1 protein level and related molecular mechanism have not been studied by us, but this does not hamper the value of our research. In the near future, we will focus on the above issues.

Reference:

[49] Miao ZF, Zhao TT, Wang ZN, Xu YY, Mao XY, Wu JH, et al. Influence of different hypoxia models on metastatic potential of SGC-7901 gastric cancer cells. Tumour Biol. 2014; 35(7): 6801-6808. doi: 10.1007/s13277-014-1928-7.

7.Comment: The introduction part of this manuscript does not clearly describe the background of current research and also does not highlight the significance of this research.

7.Reply: Thanks for pointing out our problems, we have added and refined the research background and significance in the introduction, as follows:

"Gentiopicroside is one of the important iridoid components and the main active component in Gentiaceae. Recent studies have confirmed that gentiopicroside has various pharmacological effects such as inhibiting inflammation [9], anti-apoptosis [10] and inhibiting tumor cell proliferation [11]. It can be seen that gentiopicroside is an effective biomolecule with many important biological activities and therapeutic significance. Studies have shown that gentiopicroside has good application potential for gastric mucosal injury [12] and gastric cancer [13], while necrotic apoptosis can be activated in inflammatory diseases and cancer [14]. HIF-1 has been recognized as an important anticancer drug target [15]. Accumulating evidence indicates that HIF-1α may be a potential driver of Hypoxia-induced HIF-1α/lncRNA-PMAN inhibits ferroptosis by promoting the cytoplasmic translocation of ELAVL1 in peritoneal dissemination from gastric cancer in gastric cancer[16]. The activation of HIF-1α can also promote necrotic apoptosis [17]. Therefore, we reasoned that gentiopicroside can induce necrotic apoptosis of gastric cancer through HIF-1 signaling pathway."

Reviewer#3:

1.Comment: While the study does a good job at identifying interactions and effects, deeper mechanistic insights could be achieved. For instance, it could further investigate how gentiopicroside influences other cellular pathways that might be interacting with the necroptosis pathway. While SGC7901 cells are a valid model for gastric cancer, including additional gastric cancer cell lines could help validate the findings across different genetic backgrounds and enhance the generalizability of the results.

1.Reply: Thank you for your comment. Your comments are very helpful to us. We are aware of this, relevant content has been added to the discussion, as follows:

"This study does have some limitations. This is a preliminary study on gentiopicroside participates in the necroptosis of gastric cancer cells. In the present study, we have only studied SGC-7901 cells. Some studies indicate that the level of HIF-1 protein in SGC-7901 cel

---

## [Decision Letter · Decision Letter 1]

16 Sep 2024

Gentiopicroside-Induced Gastric Cancer Necroptosis via the HIF-1 Signaling Pathway : a study involving molecular docking and experimental validation

PONE-D-24-22481R1

Dear Dr. Fan,

We’re pleased to inform you that your manuscript has been judged scientifically suitable for publication and will be formally accepted for publication once it meets all outstanding technical requirements.

Kind regards,

Jian Hao

Academic Editor

PLOS ONE

Additional Editor Comments (optional):

Reviewers' comments:

Reviewer's Responses to Questions

**Comments to the Author**

1. If the authors have adequately addressed your comments raised in a previous round of review and you feel that this manuscript is now acceptable for publication, you may indicate that here to bypass the “Comments to the Author” section, enter your conflict of interest statement in the “Confidential to Editor” section, and submit your "Accept" recommendation.

Reviewer #1: All comments have been addressed

Reviewer #2: All comments have been addressed

2. Is the manuscript technically sound, and do the data support the conclusions?

Reviewer #1: Yes

Reviewer #2: Yes

3. Has the statistical analysis been performed appropriately and rigorously? 

Reviewer #1: Yes

Reviewer #2: Yes

4. Have the authors made all data underlying the findings in their manuscript fully available?

Reviewer #1: Yes

Reviewer #2: Yes

5. Is the manuscript presented in an intelligible fashion and written in standard English?

Reviewer #1: Yes

Reviewer #2: Yes

6. Review Comments to the Author

Reviewer #1: The manuscriot has been better than the 1st edition. And this edition is recommended to be accepted by the journal Plos One except for a gentle tips: the formation "Fig 1a" should be revised as "Fig 1A" according to the writing habits, and the same to the pictures of Figure 1.

Reviewer #2: Reviewer believes that the authors of this study have basically addressed each reviewer's questions about the original study. The manuscript is ready for publication.

7. PLOS authors have the option to publish the peer review history of their article (what does this mean?). If published, this will include your full peer review and any attached files.

Reviewer #1: No

Reviewer #2: No

---

## [Editor Report · Acceptance letter]

26 Sep 2024

PONE-D-24-22481R1 

PLOS ONE

Dear Dr. Fan, 

I'm pleased to inform you that your manuscript has been deemed suitable for publication in PLOS ONE. Congratulations! Your manuscript is now being handed over to our production team.

Kind regards, 

on behalf of

Dr. Jian Hao 

Academic Editor

PLOS ONE